# Long-Term Health Symptoms and Sequelae Following SARS-CoV-2 Infection: An Evidence Map

**DOI:** 10.3390/ijerph19169915

**Published:** 2022-08-11

**Authors:** Juan Victor Ariel Franco, Luis Ignacio Garegnani, Gisela Viviana Oltra, Maria-Inti Metzendorf, Leonel Fabrizio Trivisonno, Nadia Sgarbossa, Denise Ducks, Katharina Heldt, Rebekka Mumm, Benjamin Barnes, Christa Scheidt-Nave

**Affiliations:** 1Institute of General Practice, Medical Faculty, Heinrich-Heine-University Düsseldorf, 40225 Düsseldorf, Germany; 2Research Department, Instituto Universitario Hospital Italiano de Buenos Aires, Buenos Aires C1199, Argentina; 3Department of Health Science, Universidad Nacional de La Matanza, Buenos Aires B1754JEC, Argentina; 4Department of Epidemiology and Health Monitoring, Robert Koch-Institute, 13353 Berlin, Germany

**Keywords:** long-COVID-19, COVID-19, evidence map

## Abstract

Post-COVID-19 conditions, also known as ‘Long-COVID-19’, describe a longer and more complex course of illness than acute COVID-19 with no widely accepted uniform case definition. We aimed to map the available evidence on persistent symptoms and sequelae following SARS-CoV-2 in children and adults. We searched the Cochrane COVID-19 Study Register and the WHO COVID-19 Global literature on coronavirus disease database on 5 November 2021. We included longitudinal and cross-sectional studies and we extracted their characteristics, including the type of core outcomes for post-COVID-19 conditions. We included 565 studies (657 records). Most studies were uncontrolled cohort studies. The median follow-up time was 13 weeks (IQR 9 to 24). Only 72% of studies were conducted in high-income countries, 93% included unvaccinated adults with mild-to-critical disease, only 10% included children and adolescents, and less than 5% included children under the age of five. While most studies focused on health symptoms, including respiratory symptoms (71%), neurological symptoms (57%), fatigue (54%), pain (50%), mental functioning (43%), cardiovascular functioning (40%), and post-exertion symptoms (28%), cognitive function (26%), fewer studies assessed other symptoms such as overall recovery (24%), the need for rehabilitation (18%), health-related quality of life (16%), changes in work/occupation and study (10%), or survival related to long-COVID-19 (4%). There is a need for controlled cohort studies with long-term follow-up and a focus on overall recovery, health-related quality of life, and the ability to perform daily tasks. Studies need to be extended to later phases of the pandemic and countries with low resources.

## 1. Introduction

The spread of the novel coronavirus designated as Severe Acute Respiratory Syndrome Coronavirus 2 (SARS-CoV-2), triggered not only a pneumonia but also a novel systemic disease outbreak called Coronavirus Disease 2019 (COVID-19) that rapidly disseminated across the world and which is particularly severe in older adults, leading the World Health Organisation (WHO) to declare this outbreak as a pandemic on 11 March 2020 [1].

Most people infected with SARS-CoV-2 have mild disease with nonspecific symptoms, with 6% to 41% of infected individuals never developing symptoms [2]. Approximately 5% of patients with COVID-19 experience severe symptoms and become critically ill, developing respiratory failure, septic shock, multiple organ failure, or a combination of these and requiring intensive care [3]. While the acute phase of the disease gained early scientific attention, there is still limited data on long-term outcomes [4]. Long-COVID-19 describes a longer, more complex course of illness with no widely accepted uniform case definition [5,6]. The World Health Organisation (WHO) recently developed a clinical case definition of “post-COVID-19 condition”, while other agencies use different labels to capture the long-term consequences of SARS-CoV-2 infection. While the definitions vary, there seem to be two cut-off points that have reached a consensus: 4 and 12 weeks after the acute phase, differentiating persistent or more immediate complications from long-term consequences (“post-COVID-19 Condition”) (see Table 1).

As of October 2021, we had identified several systematic reviews aiming to summarise the symptoms and sequelae following SARS-CoV-2 infection following different methodological approaches [4,11,12,13,14]. These reviews helped us plan the methods for this evidence map, whereas other systematic reviews were not included in the table as they did not assess the quality of the included studies or lacked a pre-defined protocol, which significantly reduced their reliability [15,16,17].

Considering the persisting cases of COVID-19 diagnosed worldwide and its consequences on health and healthcare systems, we aimed to collect the available evidence on persistent symptoms and sequelae following SARS-CoV-2 in children and adults in an evidence map that would serve as a resource for policymaking and evidence synthesis.

## 2. Materials and Methods

We developed an evidence map following the Global Evidence Mapping Initiative (GEM) methodology [18]. The details for these methods were registered in a predefined protocol at the Open Science Framework [19], and we followed the PRISMA extension for scoping reviews [20].

### 2.1. Inclusion Criteria

Type of studies: We included observational studies (longitudinal and cross-sectional), including those embedded in randomised controlled trials. We excluded case reports, case series, and those exclusively conducted among people with sequelae or persistent symptoms.

Type of participants: We included children and adults with documented SARS-CoV-2 infection following clinical, imaging, or laboratory criteria with an assessment of symptoms or sequelae four weeks after infection, including those with asymptomatic or mildly symptomatic infection.

### 2.2. Type of Outcome Measures

We did not use the measurement of specific outcomes assessed in studies as an eligibility criterion for this map.

### 2.3. Main Outcomes

We initially defined four core outcome measures (health-related quality of life, organ-specific symptoms or sequelae, timing and duration of symptoms, and functionality) that we would modify according to the results of the new Core Outcome Set, which was published on 31 January 2022 [21]. Based on this core outcome set and the input of Long-COVID-19 Deutschland [22], we reframed the outcomes of interest of this review to the following:Cardiovascular functioning: including symptoms and conditions (i.e., chest pain, arrhythmias, heart failure, ischaemic heart disease, and postural hypotension/tachycardia);Fatigue or Exhaustion;Pain: localised pain (headache), neuropathic pain, and muscle/joint pain;Nervous system functioning: symptoms and conditions (vertigo, dizziness, and paraesthesias/numbness);Cognitive functioning: symptoms and conditions, and systematic assessments of cognitive functioning;Mental functioning: symptoms and conditions, including but not limited to depression, anxiety, post-traumatic stress disorder, and emotional distress, among others;Respiratory functioning: symptoms (e.g., dyspnoea) and conditions, and systematic assessments of respiratory function (i.e., FEV1, FVC);Post-exertion symptoms;Health-related quality of life: including measurements of physical-mental-social functioning (SF-36 or EuroQOL or other related scales);Changes in work/occupation and studies (school attendance);Survival related to long-COVID-19 (i.e., not overall survival related to infection, but to the presence of persistent or new long-term symptoms or sequelae);Recovery/duration of symptoms;Need for rehabilitation/resource use;Other complications/sequelae: metabolic, autoimmune, or others.

The first 12 outcomes match the Core Outcome Set [21] except for “9. Physical functioning, symptoms, and conditions”, which we replaced with the wider concept of “Health-related Quality of Life” (which includes physical-mental-social functioning). We added two additional outcomes that were considered relevant to informing public health policy: the need for rehabilitation/resource use and other complications, including metabolic and autoimmune sequelae. The changes in the definition of the outcomes were made prior to data extraction.

Timing of the outcomes: The WHO case definition for post-COVID-19 condition was applied to a threshold of 12 weeks [7]. Since the acute phase of COVID-19 is usually defined by the first four weeks, remittent, persistent, fluctuating, or new symptoms present between 4–12 weeks were also included as part of the long-COVID-19 definition according to NICE guideline recommendations [9].

### 2.4. Search Methods for Identification of Studies

We performed a comprehensive, systematic search with no restrictions on the language of publication or publication status. We searched the following sources from the inception of each database:
The Cochrane COVID-19 Study Register (CCSR) was searched using the Cochrane Register of Studies (https://crsweb.cochrane.org (accessed on 5 November 2021)) which included the following:
Cochrane Central Register of Controlled Trials (CENTRAL), monthly updates.MEDLINE (PubMed), daily updates;Embase.com, weekly updates;ClinicalTrials.gov (https://www.clinicaltrials.gov, daily updates;WHO International Clinical Trials Registry Platform (ICTRP);(http://www.who.int/trialsearch, weekly updates;medRxiv (https://www.medrxiv.org, weekly updates.
WHO COVID-19 Global literature on coronavirus disease database (https://search.bvsalud.org/global-literature-on-novel-coronavirus-2019-ncov (accessed on 5 November 2021)).

The CCSR is a public, continually updated database of COVID-19 study references for which six primary sources are being regularly searched. The aim of this study-based register is to support rapid and living evidence synthesis. A recent evaluation has shown its high comprehensiveness, accurate study classifications, and short publication times [23]. Therefore, we used it as our primary source and complemented it with a second comprehensive database, which comprises several primary sources.

We constructed an empirically derived search strategy [24,25,26] for the CCSR by building a gold standard from four recently published systematic reviews on long-COVID-19 [4,11,12,14]. These four reviews conducted extensive, systematic searches on several literature databases as well as Google Scholar. In total, they included 107 studies, of which 96 were estimated as relevant to our research questions by at least one out of three researchers (JVAF, LG, and GO). All 96 references could be retrieved from the CCSR.

Our information specialist (MIM) used these 96 references as the development set for our search strategy and analysed them with the text-mining tool Voyant (voyant-tools.org). The frequency and discriminatory capacity of candidate terms derived from this set were calculated, and search strings were constructed by further analysis of the collocation of these terms. During this analysis, three references had to be excluded from the development set and classified as not efficiently retrievable. These were letters with generic titles for which an abstract was not available. The final development set consisted of 93 references. Different variations of the search strategy were developed. We chose the variant yielding 97% sensitivity (retrieving 90/93 studies of the final development set).

To search the WHO COVID-19 Global literature on coronavirus disease database, a conceptual search strategy was developed (MIM) and peer-reviewed by another information specialist (KH). The search in this source was restricted to databases not included in the CCSR.

All search strategies are available in Appendix A.

We hand searched the list of references of the included studies to identify other potential eligible studies or ancillary publications.

### 2.5. Data Collection and Analysis

We used EndNote for deduplication and Covidence for study selection. Two review authors (from GO, NS, LT, and LG) independently scanned the abstract, title, or both, of the remaining records retrieved to determine which studies should be assessed further through Covidence. Two review authors (from the same group) investigated all potentially relevant records as full text, mapped records to studies, and classified studies as included studies, excluded studies, studies awaiting classification, or ongoing studies, following the criteria for each provided in the Cochrane Handbook for Systematic Reviews of Interventions [27]. We resolved any discrepancies through consensus or recourse to a third review author (JVAF). If the resolution of a disagreement was not possible, we designated the study as ‘awaiting classification’ and contacted the study authors for clarification. We documented reasons for the exclusion of studies that may have reasonably been expected to be included in the review in a ‘Characteristics of excluded studies’ table. We presented a PRISMA flow diagram showing the study selection process [28].

### 2.6. Data Extraction

We developed a dedicated data abstraction form in Google Spreadsheets (Google LLC (Mountain View, CA, USA)) that we pilot tested ahead of time with the input of all authors. For studies that fulfil the inclusion criteria, review authors (from GO, NS, LT, and LG) independently extracted the following information from a sample of 10% of the included studies to reach a consensus, and then the rest of the studies were extracted single-handedly by a researcher: bibliographic details, details of the study design (longitudinal studies, cross-sectional studies), year of the study (2020/2021), presence of a control group, sampling method and sample size, follow-up, country and World Bank Classification (high-income country or low and middle-income country), setting (community/outpatient, hospital, ICU), age of the study population (young children, school-aged children, adolescents, young adults, middle-aged adults, older adults) and subpopulation (healthcare workers, pregnant women, etc.), vaccination status, severity COVID-19 (asymptomatic, mild, moderate, severe, critical), the definition of exposure (SARS-CoV-2 infection diagnosed by laboratory, imaging or clinically), gender/sex (male, female, trans, non-binary or other), socio-economic status (defined by income or Unsatisfied Basic Needs), assessment of prognostic factors (symptom/onset, severity/infection, vaccination status, age, gender/sex, race/ethnicity, socioeconomic status, comorbidities, non-communicable diseases, immunosuppression) and outcomes (see above).

### 2.7. Data Charting

On completion of data extraction, we transferred all the data to EPPI-Reviewer [29]. We created evidence maps by sorting the evidence according to different categories in columns and rows, including segmentation of cells according to study design, follow-up, or time of measurement after initial infection, disease severity, and measured outcomes using EPPI-Mapper [30]. We created one map for adults (>18 years old) and one map for children and adolescents (<18 years old). While most of the maps are available on the RKI website (www.rki.de/post-covid-evimaps (accessed on 8 August 2022)), in which references can be inspected interactively, we present illustrative screenshots in this manuscript.

## 3. Results

The results of our systematic literature search are summarised in the PRISMA flow chart (Figure 1). Results from databases included 9568 records from the Cochrane COVID-19 study register and 285 from the WHO Global Literature on COVID-19 database. After removing duplicates, we screened 9768 studies, of which 8681 were excluded by inspection of title and abstract, and 1087 were assessed as full text, of which 281 were excluded for various reasons. We included 565 studies (657 records) in our map. We also identified 136 ongoing studies and 13 studies awaiting classification (i.e., we contacted the authors to define eligibility and we have not received a response yet).

Most studies were available as journal articles in English. As for study design, the majority were longitudinal, uncontrolled cohort studies. The median follow-up time was 13 months, albeit two-thirds of the studies had a follow-up time greater than 12 weeks. Most studies were conducted in high-income countries (see Figure 2) and none in low-income countries. Finally, most studies included unvaccinated adults (93%) with mild-to-critical diseases. See Table 2 for the full characteristics of the included studies. Only 10% of all studies included children and adolescents, and less than 5% of all studies included young children up to five years of age and 1% of all reports focused on elderly adults. Nearly half of the studies included data on prognosis according to age (43%), gender/sex (42%), and severity of infection/symptoms at onset (43%/32%). Few studies provided data on other prognostic factors.

Figure 3 is a screenshot of a section of one of our evidence maps resulting from this research. More maps are available at www.rki.de/post-covid-evimaps (accessed on 8 August 2022).

## 4. Discussion

### 4.1. Main Findings

We identified a large body of research on post-COVID-19 conditions. Most of the research included unvaccinated adults with mild-to-critical disease, had a relatively short-term follow-up, and focused on specific health symptoms rather than global functioning and quality of life, highlighting a paucity of research in children and adolescents.

Most studies included a case mix of disease severity, considering that the recruitment was primarily done in hospitals, intensive care units, and outpatient care facilities. The main focus on respiratory outcomes is possibly related to the primary form of presentation of COVID-19. The evolving understanding of the systemic effects the virus might have led to the further exploration of its effects on other organs and systems.

### 4.2. Related Research

Our preliminary exploration of existing reviews highlighted some of the problems when collating the available evidence on post-COVID-19 conditions, as these reviews vary in quality and number of included studies, mostly due to variability in their inclusion criteria. Some of these placed restrictions on the number of participants and study design [11,12,31], which could reduce the burden of classifying and extracting data from smaller, low-quality studies. However, this might limit the inclusion of studies on subpopulations, including socially vulnerable individuals. Another approach used by others was to collate the findings of existing reviews, which allows one to summarise the findings of a larger evidence base [32]. This approach, however, relies on multiple low-quality systematic reviews, which usually fail on the definition of their methods (protocol), the rigour of their search, and the ambiguity in their inclusion criteria. Nevertheless, existing reviews highlighted similar findings to our evidence map, including the lack of controlled studies, the need for prospective studies with long-term follow-up and focus on overall recovery and health-related quality of life, and a paucity of research in children and in vaccinated people [31,32,33].

Our evidence map did not aim to cover the evidence for treatments of post-COVID-19 conditions. We considered capturing data on the treatments that the patients received in each study, but there was too much variability in their reports to aggregate in the evidence map. We consider that systematic reviews focusing on a narrower review question using data from the evidence map should analyse the data on treatments and how this relates to the incidence of post-COVID-19 conditions.

### 4.3. Limitations

Our main search strategy was empirically derived and had a high sensitivity to retrieve reports on the global incidence of post-COVID-19 conditions and sequelae, but it might be less sensitive for more studies assessing a narrower scope of complications, such as those organ-specific (e.g., diabetes, autoimmune diseases, post-exertional malaise). However, the sources we searched are comprehensive and they were used in many Cochrane reviews [34,35,36,37]. Another limitation arose due to the fact that, due to a large number of studies, we had to rely on one data-extractor per study, which may lead to coding errors; nevertheless, we took precautions, including the independent data extraction for calibration of a sample of 10% of included studies and a double-check process of all included studies when entering data into the EPPI reviewer, ensuring that extracted data correctly matched the final data presented in our study [38,39]. We expect that further inspection of the included studies in subsequent systematic reviews may give feedback dynamically to the evidence map if any errors or misclassifications occurred. Finally, we faced additional challenges due to the poor reporting of included studies, and in some cases, we had to infer study design (e.g., longitudinal as those studies that had at least two timepoints for assessment) and the severity of COVID-19 (e.g., critical as those members of the study population admitted to an intensive care unit). We assumed that the outcomes reported were those that were measured and that the prognosis factors that were reported were those that were assessed. Considering the lack of study registration of most included studies, this poses challenges in the assessment of reporting biases when conducting derivative systematic reviews. Finally, mapping outcomes reported in studies to our core outcomes posed additional challenges for several reasons. For example, results on outcomes from individual studies could be mapped into several domains (for instance, “chest pain” can be categorised as a form of “pain” as well as a cardiovascular symptom). Clearly, the currently existing core outcome set provides some orientation but needs further development in order to permit harmonised outcome definitions in primary research studies [40].

## 5. Conclusions

We identified a large body of research on post-COVID-19 conditions. Most of the research included unvaccinated adults with mild-to-critical disease, had a short-term follow-up period, and focused on specific symptoms rather than overall functioning and quality of life. Our evidence maps can be used to plan and conduct more focalised systematic reviews and meta-epidemiological studies. There is a need for controlled long-term follow-up studies assessing the health impact and the determinants of post-COVID-19 conditions in later stages of the pandemic, including vaccinated children and adults and those infected with different variants of SARS-CoV-2. Closing these data gaps will help to improve the quality of primary research studies and evidence syntheses to inform clinical practice and public health policy.

## Figures and Tables

**Figure 1 ijerph-19-09915-f001:**
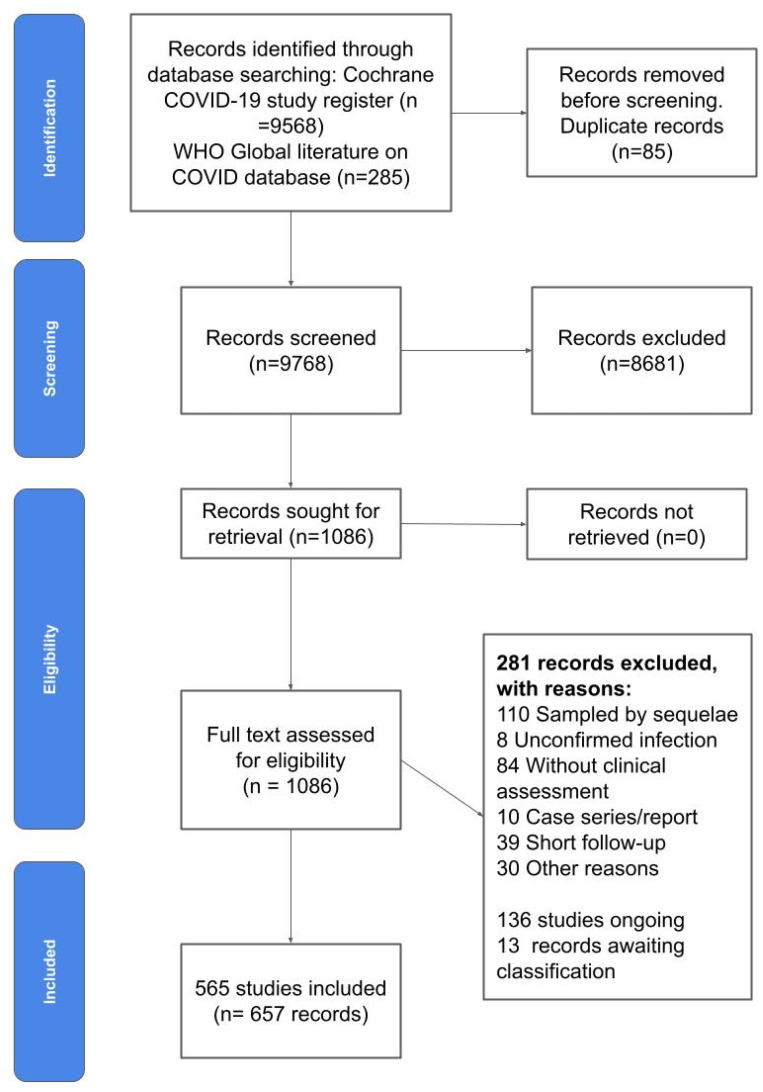
PRISMA Flow diagram.

**Figure 2 ijerph-19-09915-f002:**
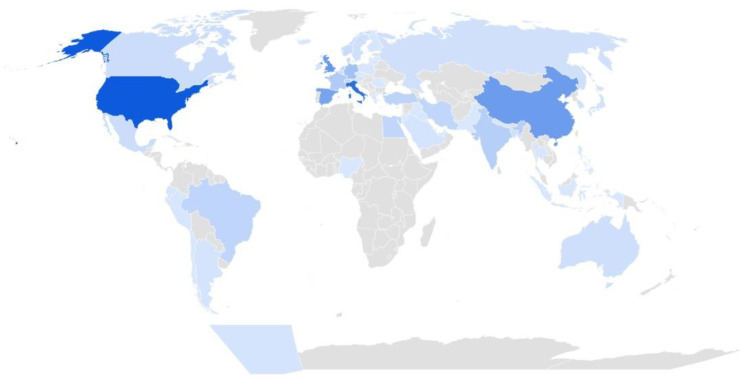
Distribution of studies per country (darker blue indicates a higher density of studies). Footnotes: Darker blue indicates a higher density of studies per country. The top 10 countries with the highest number of studies included the United States (87), Italy (75), the United Kingdom (50), China (47), Spain (42), Germany (24), France (18), India (16), Turkey (13), and Brazil (12).

**Figure 3 ijerph-19-09915-f003:**
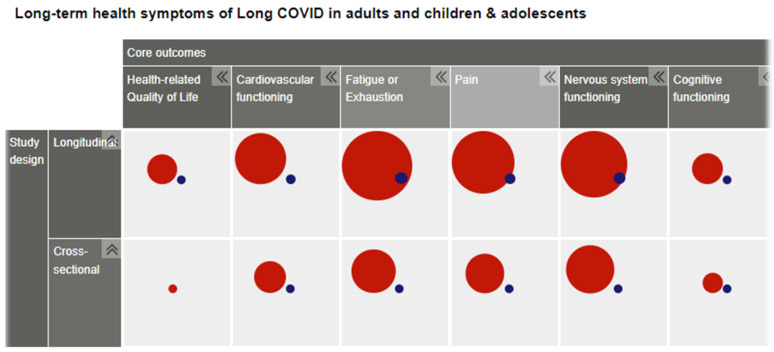
Screenshot (fragment) of an evidence map sorted out by study design, outcomes, and subgroup of adults (red) or children (blue).

**Table 1 ijerph-19-09915-t001:** Various definitions for the longer-term consequences of SARS-CoV-2 infection.

Organisation	Definition (Elements)
WHO [7]	*Post-COVID-19 conditions* Adults with a history of probable or confirmed SARS-CoV-2 infection ≥3 months from the onset of COVID-19, ≥2 months duration It cannot be explained by an alternative diagnosis Clustering of symptoms (fatigue, shortness of breath, and others) Impact on everyday functioning Symptoms may be new or persistent after recovery, fluctuate or relapse.
CDC [8] *USA*	*Post-COVID-19 conditions:* new, returning, or ongoing health problems four or more weeks after COVID-19 Even people who did not have COVID-19 symptoms initially These post-COVID-19 conditions may also be known as long COVID-19, long-haul COVID-19, post-acute COVID-19, long-term effects of COVID-19, or chronic COVID-19.
NICE [9] *UK*	*Ongoing symptomatic COVID-19:* signs and symptoms of COVID-19 from 4 to 12 weeks. *Post-COVID-19 syndrome:* signs and symptoms that develop during or after an infection consistent with COVID-19, continue for more than 12 weeks and are not explained by an alternative diagnosis. Long-COVID-19 includes ongoing symptoms and post-COVID-19 syndrome.
AWMF [10] *Germany*	New or persistent symptoms after the acute COVID-19 phase (>4 weeks) Health impairment Worsening of a pre-existing underlying disease.

**Table 2 ijerph-19-09915-t002:** Characteristics of included studies.

Characteristics	Proportion
Publication type	
Preprint	39/565 (6.90%)
Journal article	421/565 (74.51%)
Abstract	66/565 (11.68%)
Research letter/brief report	39/565 (6.90%)
Language	
English	555/565 (98.23%)
Other	10/565 (1.77%)
**Study design**	
Cross-sectional	142/565 (25.12%)
Longitudinal	422/565 (74.69%)
With a control group	83/565 (15%)
Follow-up ≥ 12 weeks	377/565 (66.73%)
Median sample size (interquartile range)	134 participants (73 to 397)
Median follow up (interquartile range)	13 weeks (9 to 24)
**Setting**	
Country	
High income	410/565 (72.57%)
Upper middle income	96/565 (17.17%)
Lower middle income	50/565 (8.85%)
Low income	0/565 (0%)
Recruitment	
Community/contact tracing	120/565 (21.24%)
Outpatient	194/565 (34.34%)
Hospital	327/565 (57.88%)
ICU	192/565 (33.98%)
**Population**	
Children	55/565 (9.73%)
Aged 0–5	26/565 (4.60%)
Aged 6–11	36/565 (6.37%)
Aged 11–18	50/565 (8.85%)
Adults	528/565 (93.45%)
Only elderly adults	7/565 (1%)
Subpopulation	
Healthcare workers	22/565 (3.89%)
Pregnant persons	1/565 (0.18%)
Socially vulnerable	15/565 (2.65%)
Chronic conditions	116/565 (20.53%)
Vaccinated	4/565 (0.71%)
Severity	
Asymptomatic	111/565 (19.65%)
Mild	306/565 (54.16%)
Moderate	346/565 (61.24%)
Severe	353/565 (62.48%)
Critical	311/565 (55.04%)
**Prognostic factors**	
Symptoms/onset	182/565 (32.21%)
Severity/infection	244/565 (43.19%)
Vaccination status	3/565 (0.53%)
Age	243/565 (43.01%)
Gender/sex	238/565 (42.12%)
Race/ethnicity	48/565 (8.50%)
Socio-economic status	43/565 (7.61%)
Comorbidities	200/565 (35.40%)
Non-Communicable Diseases	131/565 (23.19%)
Immunosuppression	24/565 (4.25%)
**Outcomes**	
Cardiovascular functioning	227/565 (40.18%)
Fatigue or Exhaustion	307/565 (54.34%)
Pain	281/565 (49.73%)
Nervous system functioning	324/565 (57.35%)
Cognitive functioning	146/565 (25.84%)
Mental functioning	241/565 (42.65%)
Respiratory functioning	401/565 (70.97%)
Post-exertion symptoms	156/565 (27.61%)
Health-related Quality of Life	92/565 (16.28%)
Changes in work/occupation and study	57/565 (10.09%)
Survival related to long-COVID-19	25/565 (4.42%)
Recovery/duration of symptoms	135/565 (23.89%)
Need for rehabilitation/resource use	101/565 (17.88%)
Other complications/sequelae	271/565 (47.96%)

## Data Availability

All the data from this research are available at the OSF website (https://osf.io/b7dwy/ (accessed on 8 August 2022)) and the RKI website (www.rki.de/post-covid-evimaps (accessed on 8 August 2022)).

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
