# Peer review of "Long-Term Health Symptoms and Sequelae Following SARS-CoV-2 Infection: An Evidence Map"

_ijerph, 2022, doi:10.3390/ijerph19169915_

Round 1
Reviewer 1 Report
TYhe authors analyzed what was reported in the literature on the so-called "post-covid", ie the symptoms reported by patients after a sars-cov2 infection. The results are interesting, they can provide useful information on post-covid management. The analysis and statistics methodology is rigorous and correct, the conclusions are consistent with the results.
Author Response
Response: Thank you so much for the positive feedback.
Reviewer 2 Report
Thanks for the opportunity to review this paper. In the article entitled “Long-term health symptoms and sequelae following SARS-CoV-2 infection: an evidence map” authors evaluated the available evidence on persistent symptoms and sequelae following SARS-CoV-2.The authors, tried to present and consider all parameters in their analyses and performed the appropriate statistical analyses. This study could be of interest and of clinical relevance for the readers of IJERPH The paper is clearly written.
Specific comments:
1. This study had some limitations due to lack of an accepted uniform case definition and to variability in inclusion criteria above all. However, limitations were clearly described by authors.
2. If possible, it would be of interest to have some information on pharmacological treatment.
Author Response
- This study had some limitations due to lack of an accepted uniform case definition and to variability in inclusion criteria above all. However, limitations were clearly described by authors.
Response: Thank you so much for the positive feedback.
- If possible, it would be of interest to have some information on pharmacological treatment.
Response: We added to the discussion (Related Research section)
“Our evidence map did not aim to cover the evidence for treatments of pot Covid-19 conditions. We considered capturing data on the treatments that the patients received in each study but there was too much variability in their reports to aggregate in the evidence map. We consider that systematic reviews focusing on a narrower review question using data from the evidence map should analyse the data on treatments and how this relates to the incidence of post Covid-19 conditions.”
Reviewer 3 Report
In the report “Long-term health symptoms and sequelae following SARS- 2
CoV-2 infection: an evidence map”, the author performed a data analysis study on COVID19-related literatures from different databases. With further data extraction and screening of the validated literatures, the author included 565 studies in their evidence map which presents the statistic features on Long-Covid19 groups. In a conclusion, the author summarized the characteristics of included literatures and suggest that more controlled cohort studies should be performed regarding long-term covid19 groups.
The work described in the manuscript is very interesting. It provides a research overview of current studies on long-covid19 groups through a systematic search approach. This report may provide knowledge and insights for future research. The description of research methods and results is clear and sufficient. Overall, this study is well-designed, with solid data that answers the author’s scientific question.
I have a few suggestions for minor changes:
1. In table 2, population group. A more precise classification of age groups in adults could provide more information to characterize a more vulnerable group to Long-Covid.
2. Figure 3 is of low quality.
3. More words regarding research results should be added in the discussion part. The interpretation of the statistical results will help readers to understand the research's significance. Any guess why respiratory functioning symptoms have a higher rate? Why statistics results of mild, moderate, severe, and critical groups are so close? etc.
Author Response
- In table 2, population group. A more precise classification of age groups in adults could provide more information to characterize a more vulnerable group to Long-Covid.
Response: Thank you for this suggestion. We have added the number of studies that only included elderly adults as defined by study authors in Table 2.
In the main text we added “Only 10% of all studies included children and adolescents, and less than 5% of all studies included young children up to five years and 1% reports focused on elderly adults.”
- Figure 3 is of low quality.
Response: Thanks for picking this up. We have uploaded a higher-resolution image with the updated data from the website. The higher resolution can be explored in interactive maps from the website.
- More words regarding research results should be added in the discussion part. The interpretation of the statistical results will help readers to understand the research's significance. Any guess why respiratory functioning symptoms have a higher rate? Why statistics results of mild, moderate, severe, and critical groups are so close? Etc.
Response: Thank you for this suggestion. We have added in the Summary of main results a preliminary interpretation of the findings, understanding that the full analysis of the included studies would pertain to subsequent systematic reviews:
“Most of the studies included a case mix of disease severity, considering that the recruitment was primarily done in hospitals, intensive care units and outpatient care facilities. The main focus on respiratory outcomes is possibly related to the primary form of presentation of Covid-19. The evolving understanding of the systemic effects of the virus might have led to the further exploration of its effects on other organs and systems.”